## [Peer Review File · Molecular Systems Biology]

Pooled single-cell screen in colorectal cancer defines transcriptional modules linked to oncogenes

Viola Hollek, Francisca Boehning, Catalina Florez-Vargas, Anja Sieber, Markus Morkel, and Nils Blüthgen

Corresponding author(s): Nils Blüthgen (nils.bluthgen@charite.de)

Review Timeline:

Submission Date:	19th Aug 25
Editorial Decision:	24th Sep 25
Revision Received:	26th Nov 25
Editorial Decision:	15th Dec 25
Revision Received:	18th Dec 25
Accepted:	23rd Dec 25

Editor: Jingyi Hou

Transaction Report:

24th Sep 2025

Manuscript Number: MSB-2025-13298-T

Title: Single-cell screen in colorectal cancer identifies transcriptional modules unlocked by oncogenes

Author: Viola Hollek

Francisca Boehning

Catalina Florez-Vargas

Anja Sieber

Markus Morkel

Nils Blüthgen

Dear Nils,

Thank you for submitting your manuscript to Molecular Systems Biology. We have now received reports from the three reviewers who agreed to evaluate your work. As you will see from their comments below, the reviewers find the study to be of interest and are generally supportive. However, they have raised several important concerns that must be addressed in a major revision.

Without reiterating all the points from the individual reviews, we would like to highlight the following key issues that require particular attention:

- Both Reviewer #1 and Reviewer #3 expressed concerns regarding the clinical utility of the TMOs and its superiority and added value compared to other patient stratification and prognosis approaches.
- Reviewer #1 also noted that the initial genetic perturbation experiments appear somewhat disconnected from the subsequent gene expression phenotypes and clinical signatures. This conceptual link should be clearly articulated and strengthened.

All other issues need to be addressed as well. As you may already know, our editorial policy allows in principle a single round of major revision, and it is therefore essential to provide responses to the reviewers' comments that are as complete as possible. Please feel free to contact me in case you would like to discuss in further detail any of the issues raised by the reviewers.

On a more editorial level, we would ask you to address the following issues:

- Please provide a .docx formatted version of the manuscript text (including legends for main figures, EV figures and tables). Please make sure that the changes are highlighted to be clearly visible.
- Please provide individual production quality figure files as .eps, .tif, .jpg (one file per figure).
- Please provide a .docx formatted letter INCLUDING the reviewers' reports and your detailed point-by-point responses to their comments. As part of the EMBO Press transparent editorial process, the point-by-point response is part of the Review Process File (RPF), which will be published alongside your paper.
- Please note that all corresponding authors are required to supply an ORCID ID for their name upon submission of a revised manuscript.
- We replaced Supplementary Information with Expanded View (EV) Figures and Tables that are collapsible/expandable online (see examples in <http://msb.embopress.org/content/11/6/812>). A maximum of 5 EV Figures can be typeset. EV Figures should be cited as 'Figure EV1, Figure EV2' etc... in the text and their respective legends should be included in the main text after the legends of regular figures.

Additional Tables/Datasets should be labeled and referred to as Table EV1, Dataset EV1, etc. Legends have to be provided in a separate tab in case of .xls files. Alternatively, the legend can be supplied as a separate text file (README) and zipped together with the Table/Dataset file.

For the figures and tables that you do NOT wish to display as Expanded View figures, they should be bundled together with their legends in a single PDF file called *Appendix*, which should start with a short Table of Content. Each legend should be below the corresponding Figure/Table in the Appendix. Appendix figures and tables should be referred to in the main text as: "Appendix Figure S1, Appendix Figure S2, Appendix Table S1" etc. See detailed instructions regarding expanded view here: <https://www.embopress.org/page/journal/17444292/authorguide#expandedview>.

- Before submitting your revision, primary datasets (and computer code, where appropriate) produced in this study need to be deposited in an appropriate public database (see <http://msb.embopress.org/authorguide-dataavailability> <https://www.embopress.org/page/journal/17444292/authorguide#dataavailability>).

The accession numbers and database should be listed in a formal "Data Availability" section (placed after Materials & Method) that follows the model below (see also <https://www.embopress.org/page/journal/17444292/authorguide#dataavailability>). Please note that the Data Availability Section is restricted to new primary data that are part of this study.

Data availability

- RNA-Seq data: Gene Expression Omnibus GSE46843 (<https://www.ncbi.nlm.nih.gov/geo/query/acc.cgi?acc=GSE46843>)

- [data type]: [name of the resource] [accession number/identifier/doi] ([URL or identifiers.org/DATABASE:ACCESSION])

Additional information on source data and instruction on how to label the files are available

- Our journal encourages inclusion of *data citations in the reference list* to directly cite datasets that were re-used and obtained from public databases. Data citations in the article text are distinct from normal bibliographical citations and should directly link to the database records from which the data can be accessed. In the main text, data citations are formatted as follows: "Data ref: Smith et al, 2001". In the Reference list, data citations must be labeled with "[DATASET]". A data reference must provide the database name, accession number/identifiers and a resolvable link to the landing page from which the data can be accessed at the end of the reference. Further instructions are available at .

- We updated our journal's competing interests policy in January 2022 and request authors to consider both actual and perceived competing interests. Please review the policy <https://www.embopress.org/competing-interests> and update your competing interests if necessary.

Please use the heading "Disclosure statement and competing interests".

- All Materials and Methods need to be described in the main text using our 'Structured Methods' format. According to this format, the Methods section includes a Reagents and Tools Table (listing key reagents, experimental models, software and relevant equipment and including their sources and relevant identifiers) followed by a Methods and Protocols section describing the methods, ideally using a step-by-step protocol format. The aim is to facilitate adoption of the methodologies across labs.

Please download and fill our Reagents and Tools Table template (.docx), which you can find in our author guidelines: <https://www.embopress.org/page/journal/17444292/authorguide#structuredmethods>.

An example of a Method paper with Structured Methods can be found here: <https://www.embopress.org/doi/10.15252/msb.20178071>.

-Regarding data quantification:

Please ensure to specify the name of the statistical test used to generate error bars and P values, the number (n) of independent experiments (please specify technical or biological replicates) underlying each data point and the test used to calculate p-values in each figure legend. Discussion of statistical methodology can be reported in the materials and methods section, but figure legends should contain a basic description of n, P and the test applied.

Graphs must include a description of the bars and the error bars (s.d., s.e.m.).

- Please provide a "standfirst text" summarizing the study in one or two sentences (approximately 250 characters, including space), three to four "bullet points" highlighting the main findings and a "synopsis image" (550px width and 400-600 px height, PNG format) to highlight the paper on our homepage.

Here are a couple of examples:

<https://www.embopress.org/doi/10.15252/msb.20199356>

<https://www.embopress.org/doi/10.15252/msb.20209475>

<https://www.embopress.org/doi/10.15252/msb.209495>

When you resubmit your manuscript, please download our CHECKLIST (<https://www.embopress.org/pb-assets/embosite/EMBO%20Press%20Author%20Checklist-1642513524327.xlsx>) and include the completed form in your submission.

Please note that the Author Checklist will be published alongside the paper as part of the transparent process (<https://www.embopress.org/page/journal/17444292/authorguide#transparentprocess>).

If you feel you can satisfactorily deal with these points and those listed by the referees, you may wish to submit a revised version of your manuscript. Please attach a covering letter giving details of the way in which you have handled each of the points raised by the referees. A revised manuscript will be once again subject to review and you probably understand that we can give you no

guarantee at this stage that the eventual outcome will be favorable.

I look forward to receiving your revised manuscript soon.

Kind regards,
Jingyi

Jingyi Hou, PhD
Senior Editor
Molecular Systems Biology

We realize that it is difficult to revise to a specific deadline. In the interest of protecting the conceptual advance provided by the work, we recommend a revision within 3 months (23rd Dec 2025). Please discuss the revision progress ahead of this time with the editor if you require more time to complete the revisions. Use the link below to submit your revision:

Link Unavailable

*** PLEASE NOTE *** As part of the EMBO Press transparent editorial process initiative (see our Editorial at <https://dx.doi.org/10.1038/msb.2010.72>), Molecular Systems Biology publishes online a Review Process File with each accepted manuscripts. This file will be published in conjunction with your paper and will include the anonymous referee reports, your point-by-point response and all pertinent correspondence relating to the manuscript. If you do NOT want this File to be published, please inform the editorial office at contact@molsystbiol.org within 14 days upon receipt of the present letter.

Reviewer #1:

Summary

The manuscript "Pooled single-cell screening in colorectal cancer identifies transcriptional modules of clinical relevance unlocked by oncogenes" by Hollek and colleagues present the generation and utility of a resource linking oncogene expression and transcriptional response. The authors aim to represent the oncogenic spectrum of colorectal cancer (CRC) by selecting and expressing 50 transgenes in five different CRC cell lines followed by single cell mRNA sequencing. They then train a conditional variational autoencoder to define 10 oncogene-driven transcriptional modules they refer to as TMOs. Those TMOs reflect different biological programs and partially reflect the genotype of some of the CRC cell lines used. The authors use those TMOs to stratify CRC patients and successfully predict clinical outcome in certain cases.

General remarks

The presented data set covering expression profiles of 100,000 single cells with different genetic backgrounds. In more technical terms, the study addresses an important area of single cell biology: the investigation of perturbation effects. Here, perturbations are caused genetically either through the different make-up of cell lines used or the transgenes introduced into those cell lines. The authors motivate both the genetic make-up of the five cell lines used as well as the transgenes well based on challenges when leveraging omics data to understand CRC. However, the presentation of the data and the exploration seems to be largely detached from this initial motivation of the study. This starts by a shallow exploration of data quality, followed by a weak attempt to connect the transgenes to expression profiles. In the next step, both the motivation for and the description of the exact set-up of the autoencoder to define the TMOs is poor. The in my understanding main results from the study are only partially conclusive. First, the authors only anecdotally connect the genetic perturbation to the expression phenotype. Second, whether the TMOs enable clinical prognostic performance better than existing transcription signature-based methods is not demonstrated. Overall, the utility of the data seems promising but major questions regarding the quality of the data remain. I believe that this work is of high interest to a wide audience and given that those fundamental concerns can be clarified or addressed, I would support its publication at Molecular Systems Biology.

Major comments

1. Reproducibility and data quality. This study will likely achieve its biggest impact as data resource - assuming sufficient data quality. However, how reproducible the data is remains unclear. The authors should go beyond using an already somewhat arbitrary data illustration such as UMAP to illustrate reproducibility. I would at least expect correlation analyses between the biological replicates. More specifically, the authors need to show that expression signatures that are unique for specific

transgenes can be reproduced (are truly specific for those transgenes). A better illustration of whether all transgenes provide useful signal in all cell lines is also required.

2. Genotype-to-phenotype association. The authors motivate their study through a lack of causal connection between known genetic CRC drivers clinically useful markers. They even claim that they "...establish mechanistic links between oncogene activity and CRC phenotypes" in their abstract. However, the manuscript currently lacks a serious attempt to connect the genetic make-up of the carefully selected cell lines and transgenes to the transcriptional signatures measured in the study. It remains unclear if cell line and transgene selection was necessary and sufficient for establishing clinically useful signatures.

3. Clinical utility. The study should demonstrate that the clinical utility (patient stratification, prognosis) of their TMOs is superior to alternative omics data.

Minor comments

1. Please describe the data set better. For instance, for how many genes do you measure acceptably high expression levels?
2. Please better explain Figure 2C. For instance, what are error bars? How to the average cell counts per variant (x label) match the 100,000 cells sequenced?
3. Please better explain / label Figure 2E.
4. Please explain how using cell line as a batch variable preserves information driven by CRC-relevant genotypes present in those cell lines. Given the small number of cell lines (5), are you worried that there is confounding between the cell line label and specific mutations?
5. Please better explain / label Figure 3C and D.
6. The order of TMO 2 and 3 are swapped in Figure 3G.
7. Please better label Figure 4.
8. Consider replacing the TMO1-10 nomenclature by something that makes it easier to understand the manuscript. Maybe name each module by what biology it represents.

Reviewer #2:

The authors present a study to link oncogenic mutations to phenotype in the form of transcriptomic signatures. They generate scRNAseq data from a collection of oncogene variants across various genetically diverse cell lines. They identify 10 transcriptomic signatures representing different pathways/processes and show their predictive value on clinical data in terms of relapse-free survival. Overall, the manuscript was a pleasure to read, with good ideas and well described results and methodology. I only have a few minor comments:

My main concern regards the TMO signatures that are capturing different stages of the cell cycle. It is well understood that perturbation responses and their output on the transcriptome will differ depending on the cell cycle stage. I believe that people often regress out the cell cycle stage effect in such analyses for that reason. Do the authors think this might be an issue here and if not can they add a clarification as to why?

As the authors also show in their analysis the genetic background does have an effect on the transcriptional output. Therefore using Progeny as a ground truth for which processes are active suffers from the limitation that the data Progeny is based on was only acquired from one specific cell line, and probably in most cases not even from colorectal cancers. There is not much to do with this, but I would highlight this limitation in the discussion for that part of the analysis.

Another thing that can be briefly mentioned in the discussion is that the microenvironment will also have an effect on the phenotypic output of oncogenic mutations. It can be added as a future outlook perhaps, so that readers don't leave with the impression that these signatures are now the holy grail on what one expects as an output of a specific mutation.

Reviewer #3:

The study presents a well-designed pooled single-cell perturbation screen across five colorectal cancer (CRC) cell lines that defines ten oncogene-unlocked transcriptional modules (TMOs) and demonstrates their prognostic value. TMOs reflect key cancer processes like plasticity, inflammation, stress, and EMT, which help stratify colorectal cancer patients by risk and improve prognosis beyond current classifications. The experimental scale, methodological integration (barcoded gain-of-function library, scRNA-seq, PROGENY/GSEA, orthogonal validation, and multi-cohort survival analyses), and clinical anchoring collectively make this work timely and impactful.

Major points

1) A primary concern regards the treatment of cell lines as "batches" during module learning with the linear-decoder scVI model, which may obscure biological heterogeneity relevant to oncogene-context dependencies. Typically, the "batch" factor is intended to represent technical variation (e.g., processing batches or sequencing runs), not biological differences such as distinct cell lines. Treating cell lines as batches risks removing real biological variability inherent to each cell line, thus confounding biological signal and batch correction. Additional analyses with reduced batch correction or cell line-stratified models are recommended to

confirm TMO robustness.

2) Module stability and rank selection: while the Jaccard-based robustness criterion supports $k=10$, it would be valuable to report stability across random seeds, alternative latent distributions, and integration strategies to ensure reproducibility of TMO.

3) Benchmarking the prognostic capacity of TMOs against established CRC signatures beyond CMS/iCMS would clarify incremental clinical utility.

Minor points

A more smooth and detailed introduction of transcriptional modules in the first paragraph of the second result section "Identification and functional characterisation of modules spanning the phenotypic space" is needed.

We thank all three reviewers for their positive and constructive comments, which we address in full below. The original review is in black, our response in blue.

Reviewer #1:

Summary

The manuscript "Pooled single-cell screening in colorectal cancer identifies transcriptional modules of clinical relevance unlocked by oncogenes" by Hollek and colleagues present the generation and utility of a resource linking oncogene expression and transcriptional response. The authors aim to represent the oncogenic spectrum of colorectal cancer (CRC) by selecting and expressing 50 transgenes in five different CRC cell lines followed by single cell mRNA sequencing. They then train a conditional variational autoencoder to define 10 oncogene-driven transcriptional modules they refer to as TMOs. Those TMOs reflect different biological programs and partially reflect the genotype of some of the CRC cell lines used. The authors use those TMOs to stratify CRC patients and successfully predict clinical outcome in certain cases.

General remarks

The presented data set covering expression profiles of 100,000 single cells with different genetic backgrounds. In more technical terms, the study addresses an important area of single cell biology: the investigation of perturbation effects. Here, perturbations are caused genetically either through the different make-up of cell lines used or the transgenes introduced into those cell lines. The authors motivate both the genetic make-up of the five cell lines used as well as the transgenes well based on challenges when leveraging omics data to understand CRC.

However, the presentation of the data and the exploration seems to be largely detached from this initial motivation of the study. This starts by a shallow exploration of data quality, followed by a weak attempt to connect the transgenes to expression profiles. In the next step, both the motivation for and the description of the exact set-up of the autoencoder to define the TMOs is poor. The in my understanding main results from the study are only partially conclusive. First, the authors only anecdotally connect the genetic perturbation to the expression phenotype. Second, whether the TMOs enable clinical prognostic performance better than existing transcription signature-based methods is not demonstrated. Overall, the utility of the data seems promising but major questions regarding the quality of the data remain. I believe that this work is of high interest to a wide audience and given that those fundamental concerns can be clarified or addressed, I would support its publication at Molecular Systems Biology.

We thank the reviewer for the detailed and positive assessment of our manuscript.

Major comments

1. Reproducibility and data quality. This study will likely achieve its biggest impact as data resource - assuming sufficient data quality. However, how reproducible the data is remains unclear. The authors should go beyond using an already somewhat arbitrary data illustration such as UMAP to illustrate reproducibility. I would at least expect correlation analyses between the biological replicates. More specifically, the authors need to show that expression signatures that are unique for specific transgenes can be reproduced (are truly specific for those transgenes). A better illustration of whether all transgenes provide useful signal in all cell lines is also required.

We fully agree that data quality metrics are important. To investigate data quality in greater detail, we now include an extended set of quality parameters. These plots show distributions of sequencing depth (nCount, nFeature), and percentage of mitochondrial reads (new Fig. EV1A). Also, we would like to refer the reviewer to Fig. EV1B (previously Suppl. Fig 2A), which shows oncogene overexpression per cell line. We also show that effect sizes (measured as E-distance) of transcriptome deregulation are an inherent quality of the oncogenes, and not merely due to overexpression strength (Appendix Fig. 1, formerly Supplementary Fig 2B).

Furthermore, to investigate reproducibility, we have now compared the independent experimental replicates concerning the module (TMO) embeddings. Both replicates show a high correlation across modules and cell lines. We spot a slight shift between replicates for the cell cycle-related TMOs 3, 4 and 5 in cell line SW480, indicating that this line might have been harvested at somewhat different cell densities between the replicates. The concordance is equally high using either the unfiltered module signatures, or the filtered ones (excluding stromal and immune-centered genes, as used for cohort analyses). These quality controls are now shown as new Appendix Figure 4, as the filtering becomes relevant when we attempt to analyze CRC cohorts.

2. Genotype-to-phenotype association. The authors motivate their study through a lack of causal connection between known genetic CRC drivers clinically useful markers. They even claim that they "...establish mechanistic links between oncogene activity and CRC phenotypes" in their abstract. However, the manuscript currently lacks a serious attempt to connect the genetic make-up of the carefully selected cell lines and transgenes to the transcriptional signatures measured in the study. It remains unclear if cell line and transgene selection was necessary and sufficient for establishing clinically useful signatures.

We agree that it would be very interesting to investigate not only the effect of drivers alone (represented by our transgenes), but also interactions between the genetic background of the cell lines and the exogenous drivers. In the current manuscript, genotype-to-phenotype associations

are explored in different chapters. For instance, the differential engagement of TMOs per oncogene and cell line is shown in Fig. 4C, and Fig. 5 provides an in-depth look into MAPK-related signal transduction and gene expression.

That said, we agree with the reviewer that the mechanistic interplay between the mutational background of the cell lines and the introduced oncogenes cannot be explored in this manuscript to a point that we can draw definitive conclusions. This limitation arises from the relatively small number of cell lines under investigation in conjunction with their complex genetic heterogeneity. We now refer to specific genotype-transgene interactions in the results (page 8, first paragraph) and discuss limitations due to low numbers of cell line models in the last paragraph of the discussion.

To formally address the question whether a representative cell line selection was necessary, we have now also performed additional scVI analyses using data from each individual cell line that are described below in our answer to Reviewer #3, major point #1. Briefly, most programs can be derived from all or several cell lines individually (see Appendix Fig. 2A).

3. Clinical utility. The study should demonstrate that the clinical utility (patient stratification, prognosis) of their TMOs is superior to alternative omics data.

It is of note that omics data such as transcriptomes are currently not used for clinical decision making in CRC. Indeed, there is a lack of transcriptome data for studies exploring targeted or immunotherapy response. Therefore, we are also limited in our manuscript to prognosis of patient survival and cannot assess prediction of (targeted) therapy response, which would have a much greater clinical utility.

In our original manuscript, we have already compared our TMO system to CMS and iCMS subtypes that are widely recognized but have also not reached clinical adoption. To further address the point of clinical utility, we have now investigated recurring oncogenic mutations, which are derived as part of standard molecular pathology. As expected, these do not meet statistical standards for prognosis, as shown below. Furthermore, we screened the literature for other gene expression signatures with potential or proposed clinical utility, as shown below in answer to rev#3. Briefly, we identified one signature derived from a mouse model of minimal residual disease, in conjunction with human cohort data, that overlaps with our TMOs 1,2 and 10 and likewise has prognostic value (see new Fig. EV4).

Reviewer figure 1: Prognostic value of common oncogenic variants (with $n > 4$, n of patients in brackets), Marisa et al. cohort. No mutation meets criteria for statistical significance.

Minor comments

1. Please describe the data set better. For instance, for how many genes do you measure acceptably high expression levels?

As outlined above, we now present standard quality parameters for single cell RNA seq in new Fig. EV1A.

2. Please better explain Figure 2C. For instance, what are error bars? How to the average cell counts per variant (x label) match the 100,000 cells sequenced?

We have modified the legend for improved clarity.

3. Please better explain / label Figure 2E.

We have also reordered the heatmap and modified the legend for improved clarity.

4. Please explain how using cell line as a batch variable preserves information driven by CRC-relevant genotypes present in those cell lines. Given the small number of cell lines (5), are you worried that there is confounding between the cell line label and specific mutations?

Within the framework of scVI with linear decoder, the learned batch correction essentially represents a linear offset. Thus, batch-correction by cell lines does remove average differences in transcriptomes (which represent baseline differences). Additionally, we now applied scVI per cell line and found a large overlap between the modules learned for individual cell lines and our modules (see Appendix Fig. 2A). We have addressed this comment more in depth as part of the response to reviewer #3, who raised a similar concern (1st major point).

5. Please better explain / label Figure 3C and D.

We have now improved the description of Figure 3C and D in the manuscript.

6. The order of TMO 2 and 3 are swapped in Figure 3G.

We thank the reviewer for pointing out this mistake and have corrected it.

7. Please better label Figure 4.

We have added the missing labels to Fig. 4. Particularly, we added the TMO names, UMAP axis, and oncogene-pathway associations.

8. Consider replacing the TMO1-10 nomenclature by something that makes it easier to understand the manuscript. Maybe name each module by what biology it represents.

We now use the biological label more consistently throughout the figures where possible and the main text.

Reviewer #2:

The authors present a study to link oncogenic mutations to phenotype in the form of transcriptomic signatures. They generate scRNAseq data from a collection of oncogene variants across various genetically diverse cell lines. They identify 10 transcriptomic signatures representing different pathways/processes and show their predictive value on clinical data in terms of relapse-free survival. Overall, the manuscript was a pleasure to read, with good ideas and well described results and methodology. I only have a few minor comments:

We thank the reviewer for the positive assessment of our manuscript.

1. My main concern regards the TMO signatures that are capturing different stages of the cell cycle. It is well understood that perturbation responses and their output on the transcriptome will differ depending on the cell cycle stage. I believe that people often regress out the cell cycle stage effect in such analyses for that reason. Do the authors think this might be an issue here and if not can they add a clarification as to why?

Indeed, cell-cycle effects represent a major source of variation in single-cell transcriptomic datasets and must be carefully considered. To address this, we have now included an analysis of TMO activity for cells of different cell cycle phases. This analysis demonstrates that TMO activity is largely independent of cell cycle phase, with the exception of the cell-cycle-associated modules (TMOs 3–5) that are strongly associated with specific cell-cycle phases. The analysis has been added as new Figure EV2D.

2. As the authors also show in their analysis the genetic background does have an effect on the transcriptional output. Therefore, using Progeny as a ground truth for which processes are active suffers from the limitation that the data Progeny is based on was only acquired from one specific cell line, and probably in most cases not even from colorectal cancers. There is not much to do with this, but I would highlight this limitation in the discussion for that part of the analysis.

We respectfully disagree with this point. The PROGENy models were learned from data across many independent cell line models, integrating diverse perturbation datasets, including data from CRC cell lines. Therefore, we consider it to be one of the most robust and unbiased approaches currently available for estimating pathway activity from transcriptomic data across different cell lines.

3. Another thing that can be briefly mentioned in the discussion is that the microenvironment will also have an effect on the phenotypic output of oncogenic mutations. It can be added as a future outlook perhaps, so that readers don't leave with the impression that these signatures are now the holy grail on what one expects as an output of a specific mutation.

We completely agree that the tumor microenvironment may have additional influence on the gene expression programs in cancer cells not covered by the TMOs, as well as predictive and prognostic value on its own, as also shown by the CMS system whose signatures are partly derived from the cancer microenvironment. Our focus was on tumor-cell intrinsic signatures, which we now explicitly state in the last paragraph of the discussion.

Reviewer #3:

The study presents a well-designed pooled single-cell perturbation screen across five colorectal cancer (CRC) cell lines that defines ten oncogene-unlocked transcriptional modules (TMOs) and demonstrates their prognostic value. TMOs reflect key cancer processes like plasticity, inflammation, stress, and EMT, which help stratify colorectal cancer patients by risk and improve prognosis beyond current classifications. The experimental scale, methodological integration (barcoded gain-of-function library, scRNA-seq, PROGENy/GSEA, orthogonal validation, and multi-cohort survival analyses), and clinical anchoring collectively make this work timely and impactful.

Thank you.

Major points

1) A primary concern regards the treatment of cell lines as "batches" during module learning with the linear-decoder scVI model, which may obscure biological heterogeneity relevant to oncogene-context dependencies. Typically, the "batch" factor is intended to represent technical variation (e.g., processing batches or sequencing runs), not biological differences such as distinct cell lines. Treating cell lines as batches risks removing real biological variability inherent to each cell line, thus confounding biological signal and batch correction. Additional analyses with reduced batch correction or cell line-stratified models are recommended to confirm TMO robustness.

We thank the reviewer for this comment and the opportunity to clarify the implications of our batch-correction approach. Firstly, the batch-correction with linear decoder only removes linear batch effects, removing only average differences in gene expression for each gene (one can think of it as the average transcriptome of the cell line). Thus, it is rather mild. Nevertheless, we agree that running the linear scVI for each cell line separately would be very interesting to understand how and if all cell lines are needed and/or integration removes interesting biology. Our new analyses show that scVI reproduces aspects of most, but not all TMOs from data of a single cell line (see new Appendix Fig. 2A). We conclude that TMO activities represent mainly, but not exclusively, cell line-overarching CRC gene expression programs.

2) Module stability and rank selection: while the Jaccard-based robustness criterion supports $k=10$, it would be valuable to report stability across random seeds, alternative latent distributions, and integration strategies to ensure reproducibility of TMO.

We agree with this suggestion and have performed the corresponding analysis running scVI 10 times with different random seeds. We observe that the TMOs are largely stable except for the cell cycle related TMO (TMO 4 and TMO 5). Also, the two MAPK-related TMOs 1 and 2 show some entanglement. This analysis is now shown in Appendix Fig. 2B.

3) Benchmarking the prognostic capacity of TMOs against established CRC signatures beyond CMS/iCMS would clarify incremental clinical utility.

It is of note that transcriptional signatures of CRC currently do not play roles in routine clinical decisions. As already pointed out above (rev #1, major point 3), our ability to test the clinical utility of the TMO signatures for potential prediction of therapy response is hampered by the lack of suitable clinical studies with associated transcriptome data (e.g. studies testing EGFR inhibition in RAF/RAF wildtype CRC, or studies assessing response to immune checkpoint blockade in MSI CRC).

That said, we have now screened the literature for further potentially predictive transcriptional signatures of CRC beyond CMS/iCMS, and we have compared them to the relevant TMO signatures. These comprise the Lgr5-centred stem cell signature related to CRC relapse from Merlos-Suarez (PMID: 21419747), a signature of CRC metaplasia derived from Joanito et al (PMID: 35773407, Fig. 7d), the EpiHR signature of metastatic recurrence related to EMP1 expression from Canellas-Socias (PMID: 36352230), and a signature of chemoresistance related to Mex3a expression from Barriga et al. (PMID: 28285904).

Of note, two signatures from Eduard Batlle's lab (EpiHR and MEX3A) show prognostic value (new Fig. EV4A), and particularly the EpiHR signature overlaps with our signatures TMO 1, TMO 2 and TMO 10 (new Fig. EV4B). Thus, adjusting for these signatures retain significant prognostic values of our TMOs, except for EpiHR, where only TMO3 remains significant. This suggests that the TMO 1,2,10 cover different aspects of the EpiHR signature (new Fig. EV4C).

Minor points

A more smooth and detailed introduction of transcriptional modules in the first paragraph of the second result section "Identification and functional characterisation of modules spanning the phenotypic space" is needed.

We have now revised the opening paragraph of the second Results section to provide a clearer and more detailed introduction of the transcriptional modules.

15th Dec 2025

Manuscript Number: MSB-2025-13298R

Title: Pooled single-cell screen in colorectal cancer defines transcriptional modules linked to oncogenes

Author: Viola Hollek

Francisca Boehning

Catalina Florez-Vargas

Anja Sieber

Markus Morkel

Nils Blüthgen

Dear Nils,

Thank you for sending us your revised manuscript. We have now heard back from the two reviewers who were asked to re-evaluate your study. As you will see, the reviewers are satisfied with the modifications made. Before we can formally accept your manuscript for publication, we would ask you to address the following editorial-level issues:

1. Please include up to five keywords in the manuscript file.
2. Remove "Authors' contribution" section from the manuscript file.
3. "Disclosure statement and competing interests" should be renamed to "Disclosure and competing interests statement".
4. Data availability: please remove the reviewer access token and make sure the datasets will be made publicly available upon the acceptance of the manuscript. The specific URL for GSE299651 dataset should be provided in the data availability statement.
5. The corresponding author's name is incomplete in the author checklist and needs to be corrected.
6. The appendix title page should include "Appendix for + manuscript title" as well as a Table of Contents with page numbers for all listed items. The nomenclature should follow "Appendix Figure Sx" and "Appendix Table Sx" throughout both the manuscript and the Appendix PDF.
7. "Supplementary_Table_1" should be renamed to "Table EV1" (with the corresponding callouts) and uploaded as Expanded View content, with the legend placed above the table in the Excel file. For the other two tables, because they are more complex, they should be updated to EV Datasets. The source file names, titles, legends, and manuscript callouts all need to be changed to Dataset EV1-EV2 instead of Data Set S2-S3 (or Supplementary_Table_2-3). The legends should be uploaded as a separate tab/sheet within each Excel file.
8. Source Data:
 - Source data for main figures should be uploaded as one (zipped) file / figure, and named as "manuscriptID_SourceDataForFigure x".
 - Source data for Figures 5F and 5G are provided, but the checklist states 5H and 5I; this needs to be clarified.
9. Please address the following issues in figure legends:
 - Please note that the exact p values are not provided in the legends of figures 5D, G, H; 6E, F,H; 7A, C, D, E; EV3, EV4 A, C.
 - Please indicate the statistical test used for data analysis in the legends of figures 6E, F; EV3, EV4 A, C
 - Please note that the box plots need to be defined in terms of minima, maxima, centre, bounds of box and whiskers, and percentile in the legends of figures 2F, EV2 D
 - Please note that information related to n is missing in the legends of figures 5D, E, G, H; 6E, F; EV1 A, EV2 D, EV4 A, C
 - Please note that the error bars are not defined in the legends of figures EV4 A, C.
 - Please note that the measure of center for the error bars needs to be defined in the legends of figures 6E, F; 7A, EV3.
10. Please remove the "Standfirst text and bullet points" from the manuscript file and upload them in a separate file.
11. Sections need to be named and the order should be corrected: Title page - Abstract - Keywords - Introduction - Results - Discussion - Methods - Data Availability - Acknowledgements -Disclosure and Competing Interests Statement - References - Figure Legends - Table(s) - Expanded View Figure Legends.

Use the following link to submit your revised paper:

Link Unavailable

Kind regards,
Jingyi

Jingyi Hou, PhD
Senior Editor
Molecular Systems Biology

If you do choose to resubmit, please click on the link below to submit the revision online before 14th Jan 2026.

Link Unavailable

We realize that it is difficult to revise to a specific deadline. In the interest of protecting the conceptual advance provided by the work, we recommend a revision within 3 months (14th Jan 2026). Please discuss the revision progress ahead of this time with the editor if you require more time to complete the revisions. Use the link below to submit your revision:

Link Unavailable

*** PLEASE NOTE *** As part of the EMBO Press transparent editorial process initiative (see our Editorial at <https://dx.doi.org/10.1038/msb.2010.72> , Molecular Systems Biology will publish online a Review Process File to accompany accepted manuscripts. When preparing your letter of response, please be aware that in the event of acceptance, your cover letter/point-by-point document will be included as part of this File, which will be available to the scientific community. More information about this initiative is available in our Instructions to Authors. If you have any questions about this initiative, please contact the editorial office (msb@embo.org).

Reviewer #1:

The authors have address my comments convincingly and the work has improved. I now believe that this work will be of great interest for a wide audience at Molecular Systems Biology.

A small technical remark: labeling the figure PDFs with their figure number (and title) in the PDF itself would improve navigation substantially.

Reviewer #3:

The authors have addressed all of my previous comments. I am satisfied with the revised manuscript and have no further concerns. I recommend the manuscript for publication.

The authors have addressed all minor editorial requests.

23rd Dec 2025

Manuscript number: MSB-2025-13298RR

Title: Pooled single-cell screen in colorectal cancer defines transcriptional modules linked to oncogenes

Dear Nils,

Thank you again for sending us your revised manuscript. We are now satisfied with the modifications made and I am pleased to inform you that your paper has been accepted for publication.

You may qualify for financial assistance for your publication charges - either via a Springer Nature fully open access agreement or an EMBO initiative. Check your eligibility: <https://link.springer.com/journal/44320/how-to-publish-with-us>

Sincerely,
Jingyi

Jingyi Hou, PhD
Senior Editor
Molecular Systems Biology

>>> Please note that it is Molecular Systems Biology policy for the transcript of the editorial process (containing referee reports and your response letter) to be published as an online supplement to each paper. If you do NOT want this, you will need to inform the Editorial Office via email immediately. More information is available here: <https://link.springer.com/partners/embo-press/editorial-policies#Peer%20review>